# Region-Specific Impact of Repeated Synthetic Cannabinoid Exposure and Withdrawal on Endocannabinoid Signaling, Gliosis, and Inflammatory Markers in the Prefrontal Cortex and Hippocampus

**DOI:** 10.3390/biom15030417

**Published:** 2025-03-14

**Authors:** Evelin Vadas, Antonio J. López-Gambero, Antonio Vargas, Miguel Rodríguez-Pozo, Patricia Rivera, Juan Decara, Antonia Serrano, Stella Martín-de-las-Heras, Fernando Rodríguez de Fonseca, Juan Suárez

**Affiliations:** 1Instituto de Investigación Biomédica de Málaga y Plataforma en Nanomedicina-IBIMA Plataforma BIONAND, 29590 Málaga, Spain; vadas.evelin@gmail.com (E.V.); antonio.lopez@ibima.eu (A.J.L.-G.); antonio.vargas@ibima.eu (A.V.); rpmiguel@uma.es (M.R.-P.); patricia.rivera@ibima.eu (P.R.); juandecara@uma.es (J.D.); antonia.serrano@ibima.eu (A.S.); smdelasheras@uma.es (S.M.-d.-l.-H.); 2Unidad Clínica de Neurología, Hospital Regional Universitario de Málaga, 29010 Málaga, Spain; 3INSERM, Neurocentre Magendie, University of Bordeaux, 33000 Bordeaux, France; 4UGC Salud Mental, Hospital Regional Universitario de Málaga, 29010 Málaga, Spain; 5Departamento de Anatomía Humana, Medicina Legal e Historia de la Ciencia, Universidad de Málaga, 29071 Málaga, Spain

**Keywords:** synthetic cannabinoids, WIN 55,212-2, HU-210, hippocampus, prefrontal cortex, gliosis, inflammation, withdrawal, toxicology, legal medicine, neuroanatomy

## Abstract

Synthetic cannabinoid use raises concerns about its neuroinflammatory effects, including molecular adaptations of the endocannabinoid system (ECS) in the brain. This study investigates the pharmacological effects of 14-day repeated intraperitoneal administration, as well as 14-day administration followed by a 7-day withdrawal period of two synthetic cannabinoids: WIN55,212-2 and HU-210. The study assessed gene expression and protein markers related to the ECS, gliosis, and inflammation in two brain regions critical for cognitive processes and memory—key components of addiction pathways—the prefrontal cortex (PFC) and the hippocampus of rats. Our findings showed that repeated WIN55,212-2 administration induced adaptations in the ECS and reduced IBA1, a glial protein marker, along with inflammatory responses likely mediated through CB2 activity. Notably, regional differences emerged in the hippocampus, where repeated administration of WIN55,212-2 and HU-210 increased IBA1 and inflammatory markers, effects unrelated to CB2 activity. Withdrawal from WIN55,212-2 in the PFC, as well as from both compounds in the hippocampus, decreased IBA1 levels. This was associated with altered protein expression of cannabinoid-synthesizing and degrading enzymes, favoring acylethanolamide synthesis. These findings highlight region-specific effects of synthetic cannabinoids on cannabinoid signaling, gliosis, and inflammation. Further research is needed to elucidate the long-term neurobiological consequences of synthetic cannabinoid use and withdrawal.

## 1. Introduction

Cannabis and synthetic cannabinoids are increasingly used recreationally, as well as medically, to treat pain, nausea, depressive symptoms, anorexia related to weight loss, tumors, tremors, and spasticity associated with multiple sclerosis, along with other neurological disorders such as epilepsy, Alzheimer’s, Parkinson’s and Huntington’s disease [1]. Synthetic cannabinoids are present in designed drugs often marketed and sold as herbal blends named K2 and Spice that are intended to mimic delta-9-tetrahydrocannabinol (THC) actions, the main psychoactive constituent of cannabis [2]. However, these synthetic cannabinoids often display a pharmacological potency and toxicity that pose tremendous risks for users and have forced international controlling measures from regulatory and law enforcement agencies which face a lack of knowledge on the potential toxicity caused by these products, and which is not usually associated with THC-like psychoactivity [3,4]. Thus, effects associated with both chronic consumption and withdrawal from the use of synthetic cannabinoids might include not only the traditional effects of chronic exposure to cannabis preparations but also additional health problems. As an example, chronic administration of natural cannabinoids leads to tolerance of acute effects such as hypoactivity, lethargy, and hypothermia. Consequently, cessation of cannabinoid use can trigger a withdrawal syndrome that includes anxiety and tachycardia. Consumption and withdrawal from synthetic cannabinoids, however, may result in a severe neurological syndrome, often accompanied by psychotic symptoms and, in some cases, even death [5,6,7].

The first synthetic cannabinoids were designed based on the structure of THC. HU-210, a classical cannabinoid, was first synthesized in 1980 and has been found in marketed cannabinoid products since 2009 [8]. Other synthetic cannabinoids have significantly different structures or hybrid combinations of non-cannabinoid and cannabinoid structures, which were later discovered and showed a wide range of cannabinoid-like properties. These include derivatives from cyclohexylphenol, eicosanoids, and aminoalkylindoles, with WIN 55,212-2 being one of the best representatives in the latter group of synthetic cannabinoids. These compounds, unlike THC, exhibit potent activity at both known cannabinoid receptors, with CB1 receptors being predominantly expressed in the brain and linked to neurophysiological functions, while CB2 receptors are more prevalent in peripheral immune cells and glial cells in the CNS [9]. The higher affinity of synthetic cannabinoids for cannabinoid receptors leads to a rapid tolerance with severe withdrawal symptoms and highlights the need to understand the mechanisms associated with cannabinoid signaling modulation induced by synthetic cannabinoid abuse.

Cortical GABAergic interneurons contain higher levels of CB1 than cortical glutamatergic neurons, whereas ablation of CB1 in glutamatergic hippocampal neurons has a more profound impact on neurotransmission compared to specific inhibition of CB1 in GABAergic neurons [10]. Although CB2 expression is mainly related to conditions of inflammation, CB2 ablation in mice is also related to impaired long-term potentiation in the hippocampus associated with long-term memory deficits and induces a schizophrenia-like phenotype [11]. However, microglial CB2 and CB1 activation is seen to counter pro-inflammatory mediators in the brain, and the absence of CB1 in specific neuronal types accelerates neuronal loss and promotes chronic neuroinflammation [12,13].

WIN 55,212-2 is a non-selective cannabinoid receptor agonist with partial agonist activity at the CB1 receptor and full agonist activity at the CB2 receptor. In acute administration, it modulates neuronal activity by inhibiting GABAergic and glutamatergic synaptic neurotransmission [14]. Repeated administration of WIN 55,212-2 induces long-lasting effects during withdrawal in the dopaminergic system associated with the psychoactive rewarding and locomotor effects [15]. HU-210 is a highly potent full agonist at both CB1 and CB2 receptors, with an extended half-life. It exerts a stronger inhibitory effect on GABAergic neurotransmission in the hippocampus compared to WIN 55.212-2 and THC [16]. Nevertheless, chronic stimulation of the endocannabinoid system with these drugs results in long-lasting sustained effects affecting GABA release [16], which might account for the enhanced risk of aberrant neurological and psychiatric manifestation derived from its uncontrolled use [6]. The mechanisms underlying the regulation of neurotransmitter release vary substantially between brain regions and, specifically, the impact of synthetic cannabinoids in the fine regulation of the endocannabinoid system in corticolimbic regions has not yet been elucidated. The intricate role of the cannabinoid system linking immune regulation and synaptic functioning makes it essential to describe the impact of synthetic cannabinoids on CB1 and CB2 in the brain regions of the prefrontal cortex (PFC) and hippocampus, which are critical for higher cognitive functions and especially vulnerable to neuroinflammation and gliosis.

Here, we aim to determine the impact of repeated administration of WIN 55,212-2 and HU-210 on the endocannabinoid system, gliosis, and the inflammatory response in the PFC and hippocampus of rats. By comparing both acute and chronic phases of cannabinoid exposure and withdrawal, we sought to clarify their impact on two regions crucial for cognitive performance with emphasis on the short and long-lasting consequences during the regime of administration and withdrawal. Understanding the molecular changes occurring during exposure to synthetic cannabinoids will provide critical insights into their potential adverse effects concerning neuroinflammatory and neurological processes, a finding essential for delimitating the framework for its potential therapeutic use.

## 2. Materials and Methods

### 2.1. Ethics Statement

All experiments were realized in compliance with the ARRIVE guidelines, approved by the Ethics and Research Committee at the Universidad de Málaga (ref. no. 2014-0009-A, date: 4 September 2014), and carried out in strict accordance with the European Communities Directive 2010/63/EU on the protection of animals used for scientific purposes and Spanish legislation (RD 53/2013 and 178/2004, Ley 32/2007 and 9/2003, and Decreto 320/2010) for the care and use of laboratory animals. Every effort was made to minimize animal suffering and to reduce the number of animals used.

### 2.2. Animals

Forty-eight male Wistar rats (Charles River Laboratories, Saint-Germain-Nuelles, France), weighting 200–250 g, were housed individually in controlled conditions: 12 h light/dark cycle (lights off at 20:00 p.m.), ambient temperature (22 ± 1 °C) humidity (40 ± 5%). Water and standard rodent chow were available ad libitum.

### 2.3. Drug Administration

WIN 55,212-2 (I-(+)-[2,3-Dihydro-5-methyl-3-(4-morpholinylmethyl)pyrrolo [1,2,3-de]-1,4-benzoxazin-6-yl]-1-naphthalenylmethanone mesylate; cat. no. 1038, Tocris, Bristol, UK), a cannabinoid receptor agonist with high affinity for CB2 (Ki = 3.3 nM) and partial agonist activity at CB1, was dissolved in a vehicle composed of saline with 5% of ethanol and 5% of emulphor and made fresh every day. WIN 55,212-2 at a dose of 2 × 2 mg/Kg/day for 14 days was administered intraperitoneally (i.p.) [17]. HU-210 ((6aR)-trans-3-(1,1′-dimethylheptyl)-6a,7,10,10a-tetrahydro-1-hydroxy-6,6-dimethyl-6H-dibenzo[b,d]pyran-9-methanol, cat. no. 0966, Tocris, Bristol, UK), a highly potent cannabinoid receptor agonist (Ki = 0.061 and 0.52 nM for human CB1 and CB2 receptors, respectively), was prepared fresh each day and dissolved in a saline/ethanol/emulphor (90:5:5 *v*/*v*/*v*) vehicle solution. HU-210 at a dose of 2 × 50 µg/Kg/day for 14 days was administered i.p. [18].

### 2.4. Experimental Design

The experimental design is shown in Figure 1. Adult rats (PND60) were randomly assigned to repeated drug administration for 14 days (REP group, *n* = 24) and drug withdrawal for 7 days after administration (WD group, n = 24). During the withdrawal period, animals were not handled or injected. Thus, rats from each group were randomly assigned to vehicle, WIN 55,212-2, or HU-210 experimental subgroups [i.p. injections: REP vehicle (n = 8), REP WIN 55,212-2 (n = 8), REP HU-210 (n = 8): no i.p. injections: WD vehicle (n = 8), WD WIN 55,212-2 (n = 8) and WD HU-210 (n = 8) subgroups].

### 2.5. Tissue Collection

All animals were sacrificed at PND74 (REP group) and PND81 (WD group) by decapitation after administration of Equitesin^®^ (3 mL/kg i.p.; chloral hydrate 2.1 g, sodium pentobarbital 0.46 g, MgSO_4_ 1.06 g, propylene glycol 21.4 mL, ethanol (90%) 5.7 mL, and H2O 3 mL) 24 h after the last drug administration (REP group) or after 7 days of drug withdrawal (WD group) in a room separate from the other experimental animals. Brains were quickly collected, immediately frozen on dry ice, and stored at −80 °C. The brains were then sliced on dry ice to obtain 1 mm thick sections using razor blades and rat brain slicer matrices. Brain samples from the prefrontal cortex (PFC) from 5.00 to 3.00 mm of Bregma and hippocampus (HC) from −2.16 to −4.20 mm of Bregma were precisely dissected out bilaterally with fine surgical instruments [19]. Brain samples were weighed and stored at −80 °C until used for mRNA and protein analysis.

### 2.6. RNA Isolation and RT-qPCR Analysis

Real-time PCR (TaqMan, ThermoFisher Scientific, Waltham, MA, USA) was used, as described previously [20], to quantify mRNA levels of relevant proteins, enzymes, and receptors involved in: (a) endocannabinoid signaling [CB1 (*Cnr1*), CB2 (*Cnr2*), and peroxisome proliferator-activated receptor-α, PPARα (*Ppara*)]; synthesis enzymes [NAPE-PLD (*Nape-pld*), DAGLα (*Dagla*), and DAGLβ (*Daglb*)]; degradation enzymes [FAAH (*Faah*) and MAGL (*Mgll*)]; (b) gliosis-related genes (*Iba1*, *Gfap*, *Mrc1*, *Fcgr2b,* and vimentin); and (c) inflammatory-related genes (*Il1β*, *Ikbkb*, *Nos2*, *Ptgs2*, and *Rela*), using specific sets of primer probes from TaqMan^®^ Gene Expression Assays (see Appendix A). Briefly, brain samples were homogenized on ice and total RNA was extracted following Trizol^®^ method according to the manufacturer’s instructions (ThermoFisher Scientific). RNA samples were isolated with an RNAeasy MinElute Cleanup Kit, including digestion with DNase I column (Qiagen, Hilden, Germany) and concentrations were quantified using a spectrophotometer to ensure ratios of absorbance at 260 to 280 nm of 1.8–2.0. After reverse transcript reaction from 1 µg of RNA (Transcriptor RT; Roche Diagnostic, Mannheim, Germany), quantitative real-time transcription polymerase chain reaction (qPCR) was performed in a CFX96TM Real-Time PCR Detection System (Bio-Rad, Hercules, CA, USA) and the FAM dye-labeled format for the TaqMan^®^ Gene Expression Assays (ThermoFisher Scientific). Melting curve analysis was performed to ensure that only a single product was amplified. Ct values of each sample were normalized to β-actin (*Actb*) mRNA levels.

### 2.7. Western Blotting

Brain tissues were homogenized and lysed in RIPA buffer (Sigma-Aldrich, St Louis, MO, USA) containing phosphatase inhibitor (Halt™ Phosphatase Inhibitor Cocktail, Thermo Fisher Scientific, Waltham, MA, USA) and nuclease (Pierce™ Universal Nuclease for Cell Lysis, Pierce, IL, USA). Aliquots of lysate from brain samples were electrophoresed in 4–12% sodium-dodecyl-sulfate-denaturing polyacrylamide gels, transferred onto nitrocellulose membranes (Bio-Rad), and controlled by Ponceau red staining. Blots were blocked with Tris-buffered saline containing 0.1% Tween 20 (TTBS) with 5% (*w*/*v*) bovine serum albumin (BSA) or non-fat milk for 2 h at 25 °C. For protein detection, each blotted membrane lane was incubated separately with the specific primary antibodies (see Appendix A), diluted in the blocking buffer, overnight at 4 °C. After extensive washing in TTBS, the membranes were incubated with the corresponding peroxidase-conjugated goat anti-rabbit antibody or the peroxidase-conjugated goat anti-mouse IgG antibody (Promega, Madison, WI, USA), diluted 1:10,000 in TTBS, for 1 h at room temperature and protected from light. Peroxidase activity was detected using an ECL system (Bio-Rad Laboratories, Alcobendas, Spain), and the chemiluminescent signal was calculated with ImageQuant Las 4000 Software (GE Healthcare Life Sciences, Barcelona, Spain). Gel-loading variabilities were normalized to γ-adaptin protein levels (see Appendix A for unedited gels and immunoblotted membranes).

### 2.8. Statistical Analysis

Data were analyzed using the GraphPad Prism 7.0 software (GraphPad Software Inc., San Diego, CA, USA). Data were represented as mean ± standard error of the mean (SEM) for each experimental subgroup (*n* = 8/subgroup). The significance of differences within and between subgroups was evaluated by using a two-way analysis of variance (ANOVA), whose factors were termed *drug* (vehicle vs. WIN 55,212-2 or HU-210) and *dependence* (for the dependence phase, and reflected in the groups of repeated administration vs. withdrawal) followed by Tukey post hoc test for multiple comparisons between two groups (*). A *p* value < 0.05 was considered statistically significant (* = *p* < 0.05, ** = *p* < 0.01, *** = *p* < 0.001).

## 3. Results

### 3.1. Repeated Administration and Withdrawal of WIN 55,212-2 and HU-210 Lead to Distinct Effects on the Endocannabinoid System in the Prefrontal Cortex

A significant dependence effect was observed in *Cnr1*, whereas *Cnr2* showed a significant interaction between dependence and drug, as well as a significant effect of dependence (the statistical results are shown in Appendix A). WIN 55,212-2 administrations significantly reduced *Cnr2* expression during repeated administration (*p* < 0.05), which was reversed during withdrawal (*p* < 0.001). These results suggest that the effect of WIN 55,212-2 on *Cnr2* mRNA levels depends on dependence status (Figure 2A). Significant dependence and drug effects were also observed in *Ppara*, whereas both *Daglb* and *Faah* showed significant interaction between dependence and drug, as well as a significant effect of dependence (Appendix A). Specifically, *Ppara* mRNA was elevated during withdrawal in the vehicle (*p* < 0.001), WIN 55,212-2 (*p* < 0.01), and HU-210 (*p* < 0.05) administrations (Figure 2A). There was a non-significant tendency to decrease *Dagla* mRNA during administration of WIN 55,212-2, and significant for *Daglb* (*p* < 0.05), while it was reversed during withdrawal (*p* < 0.05 for *Dagla*, *p* < 0.001 for *Daglb*). *Mgll* and *Faah* expression was also elevated with WIN 55,212-2 in withdrawal respect to the vehicle (*p* < 0.05 for *Mgll* and *Faah*), and also with respect to repeated administration of WIN 55,212-2 (*p* < 0.05 for *Mgll*, *p* < 0.001 for *Faah*) (Figure 2A).

Changes in mRNA expression were accompanied by significant effects of the dependence phase and drug administration in CB1 and CB2 receptors (Appendix A), where WIN 55,212-2 significantly reduced the protein levels of CB1 and CB2 during repeated administration (*p* < 0.05 for CB1, *p* < 0.01 for CB2), and repeated HU-210 administration significantly reduced CB1 protein levels (*p* < 0.01) (Figure 2B). These results confirm that synthetic cannabinoid administration decreased cannabinoid receptor expression and protein levels as a result of repeated administration, an effect consistent with the development of tolerance through downregulation of the receptor, as happens with THC [21]. A significant drug effect and interaction between the drug and the dependence phase was observed for PPARα (Appendix A), where repeated administration of WIN 55,212-2 (*p* < 0.001) and HU-210 (*p* < 0.01) significantly elevated its levels as compared to the vehicle (Figure 2B). A significant drug effect was also observed for cannabinoid-synthesizing enzymes DAGLα, DAGLβ and NAPE-PLD (Appendix A), where acute WIN 55,212-2 administration significantly reduced DAGLβ and NAPE-PLD protein levels (*p* < 0.05 for DAGLβ, *p* < 0.001 for NAPE-PLD) and HU-210 administration increased DAGLβ levels during withdrawal (*p* < 0.001) (Figure 2B). NAPE-PLD levels, however, were significantly increased after cessation of WIN 55,212-2 and HU-210 administration (*p* < 0.05 for both drugs).

Cannabinoid-degrading enzymes MAGL and FAAH also depicted significant drug effects and interaction between the drug and the dependence phase (Appendix A), with a significant effect of repeated administration of WIN 55,212-2 and HU-210 in reducing MAGL levels (*p* < 0.001 for both drugs) and increasing FAAH levels (*p* < 0.001 for WIN 55,212-2) (Figure 2B). Since MAGL is involved in 2-arachidonoylglycerol (2-AG) degradation and FAAH in anandamide (AEA) degradation, decreases in MAGL levels and increases in FAAH levels during repeated drug administration could reflect a shift towards 2-AG accumulation, probably as an attempt to overcome the receptor occupancy by exogenous cannabinoids being administered.

### 3.2. Repeated Administration and Withdrawal of WIN 55,212-2 and HU-210 Modulate Gliosis and Inflammatory Markers in the Prefrontal Cortex

We also investigated the impact of WIN 55,212-2 and HU-210 on the glial response, including microglia and astroglia, in the PFC. We found a significant effect of HU-210 withdrawal resulting in a reduction in *Gfap* and *Fcgr2b* expression (*p* < 0.05 for both markers), whereas WIN 55,212-2 significantly reduced *Rela* expression during repeated administration phase (*p* < 0.001) (Figure 3A). Significant changes in mRNA expression between repeated administration vs. dependence phases were also observed for *Mrc1*, *Il1b*, *Ikbkb*, *Nos2*, *Ptgs2*, and *Rela*, indicating that cessation of administration induced changes in the expression of inflammatory markers (Appendix A).

Regarding protein levels, we found a significant dependence effect for IBA1 but not for GFAP and a significant interaction between the drug and the dependence phase for IBA1 and GFAP (Appendix A), which was accompanied by decreased levels of IBA1 during repeated administration (*p* < 0.001) and withdrawal (*p* < 0.01) of WIN 55,212-2 (Figure 3B). We also found a significant drug effect for vimentin (Appendix A), with significantly increased protein levels during WIN 55,212-2 administration (*p* < 0.001) (Figure 3B). Moreover, we found a significant effect of the drug, the dependence phase, and an interaction between both factors for NF-κB and IKKβ (Appendix A), which was accompanied by decreased protein levels during repeated administration of WIN 55,212-2 (*p* < 0.05 for both proteins) (Figure 3B). COX2 levels were, however, increased during HU-210 withdrawal (*p* < 0.05) (Figure 3B). Our findings indicate that repeated administration and withdrawal of WIN 55,212-2 modulate immune-related markers in the PFC, consistent with the role of endocannabinoids in regulating immune responses through cannabinoid receptors [8].

### 3.3. Repeated Administration and Withdrawal of WIN 55,212-2 and HU-210 Differentially Modify the Endocannabinoid System in Hippocampus

Significant drug effects and interactions between dependence and drug were observed in *Cnr2* (Appendix A), where repeated administration of WIN 55,212-2 (*p* < 0.001) and HU-210 (*p* < 0.01) caused decreased mRNA levels (Figure 4A). There was a significant drug and dependence effect on *Ppara* (Appendix A), with a non-significant and significant (*p* < 0.05) decrease in *Ppara* expression during withdrawal of WIN 55,212-2 and HU-210, respectively (Figure 4A). The same trend was observed for *Dagla*, with a significant decrease (*p* < 0.05) in *Dagla* expression during withdrawal of both WIN 55,212-2 and HU-210 (Figure 4). A significant dependence phase effect was observed for *Mgll*, whereas a significant interaction between the drug and the dependence phase was also found in *Mgll* and *Faah* expression (Appendix A). These changes were accompanied by significantly higher expression of *Mgll* during withdrawal in the vehicle (*p* < 0.001), WIN 5,212-2 (*p* < 0.01), and HU-210 (*p* < 0.001), whereas *Faah* expression was increased during repeated administration of WIN 55,212-2 (*p* < 0.05), and decreased during withdrawal (*p* < 0.01) (Figure 4A).

Regarding protein levels, we found a specific interaction effect for CB1, and a significant dependence effect on CB2 and PPARα translated into overall increased protein levels during withdrawal (Appendix A) (Figure 4B). This was accompanied by a substantial CB1 increase during repeated administration of WIN 55,212-2 (*p* < 0.01) (Figure 4B). We also observed a specific drug effect and an interaction between the drug and the dependence phase in 2-AG-synthetizing enzymes DAGLα and DAGLβ (Appendix A). Specifically, repeated HU-210 administration significantly decreased DAGLα protein levels (*p* < 0.05), whereas withdrawal of WIN 55,212-2 caused decreased protein levels in DAGLα (*p* < 0.05) and DAGLβ (*p* < 0.05), and withdrawal of HU-210 caused a strong decrease in DAGLα levels (*p* < 0.001) (Figure 4B). These results indicate that repeated administration of WIN 55,212-2 has a mild effect on the hippocampal endocannabinoid system, whereas withdrawal of synthetic drugs caused a decrease in the protein levels of 2-AG-synthetizing enzymes in the hippocampus.

### 3.4. Repeated Administration of WIN 55,212-2 and HU-210 Increases Markers of Gliosis in the Hippocampus That Are Resolved During Withdrawal

The impact of WIN 55,212-2 and HU-210 on the glial response and inflammation markers in the hippocampus was also investigated. Altogether, we found a significant effect of the dependence phase in *Iba1*, *Gfap,* and *Mrc1* expression markers (Appendix A), with *Iba1* being decreased during withdrawal of the vehicle (*p* < 0.01) and WIN 55,212-2 (*p* < 0.01) administration, and non-significantly for HU-210 (Figure 5A). *Gfap* and *Mrc1* expression, however, increased after withdrawal of the vehicle (*p* < 0.001 for *Gfap*, *p* < 0.01 for *Mrc1*) and HU-210 (*p* < 0.001 for *Gfap*, *p* < 0.05 for *Mrc1*), whereas withdrawal of WIN 55,212-2 decreased *Gfap* and *Mrc1* expression (*p* < 0.05 for both markers) (Figure 5A).

A significant effect of the dependence phase was found for *Il1b*, *Ikbkb*, *Nos2*, *Ptgs2,* and *Rela*, with a significant interaction between the drug and the dependence phase in *Fcgr2b*, *Ikbkb*, and *Nos2*, and non-significantly for *Rela* (Appendix A). Interestingly, withdrawal of WIN 55,212-2 (*p* < 0.01) and HU-210 (*p* < 0.05) decreased *Fcgr2b* expression (Figure 5A). The same was observed for *Rela* expression with WIN 55,212-2 (*p* < 0.001) and HU-210 (*p* < 0.01) withdrawal, whereas HU-210 withdrawal also increased *Nos2* expression (*p* < 0.001) (Figure 5A).

As opposed to mRNA expression, we did observe a significant effect of the dependence phase and drug administration in IBA1 (Appendix A), with increased levels during repeated administration of WIN 55,212-2 (*p* < 0.05) and HU-210 (*p* < 0.05) and decreased levels during withdrawal (*p* < 0.01 for WIN55,212-2, *p* < 0.05 for HU-210) (Figure 5B). A significant effect of the dependence phase was also observed for GFAP (Appendix A), with decreased protein levels during withdrawal of WIN 55,212-2 (*p* < 0.05) and HU-210 (*p* < 0.01) (Figure 5B). A significant drug effect was also found for vimentin, NF-κB, and IKKβ (Appendix A), with a potent increase in protein levels induced by repeated administration of WIN 55,212-2 (*p* < 0.001 for all proteins), and with HU-210 in vimentin (*p* < 0.01) and IKKβ (*p* < 0.001) (Figure 5B). These results indicate that repeated administration of synthetic cannabinoids induces the protein expression of both gliosis and inflammatory markers that ceases or decreases during the withdrawal phase in the hippocampus.

### 3.5. Relationship Between Gliosis, Inflammation Markers, and Endocannabinoid System During Repeated Administration and Withdrawal of WIN 55,212-2 and HU-210

Given the known interactions between the endocannabinoid system and neuroimmune signaling, we investigated whether gene expression changes in gliosis and inflammatory markers were associated with alterations in cannabinoid receptors and enzymes involved in endocannabinoid metabolism. We performed Spearman correlation analyses to explore these relationships.

During repeated administration in the PFC, *Gfap, Ikbkb*, and *Nos2* expression levels showed a positive correlation with *Cnr1* expression, despite no significant changes in expression being previously observed (Appendix A) (Figure 6A). Additionally, *Ikbkb, Nos2, Ptgs2,* and *Rela* were positively associated with enzymes involved in 2-AG metabolism, including *Dagl* and *Mgll* (Appendix A) (Figure 6A). These results suggest a potential link between inflammatory signaling and endocannabinoid metabolism during cannabinoid exposure.

Following withdrawal, inflammatory responses in the PFC appeared less pronounced, with positive correlations observed between *Iba1* (which was reduced during WIN 55,212-2 and HU-210 withdrawal), *Fcgr2b*, and *Cnr2* expression. Additionally, *Ptgs2* and *Rela* were associated with *Cnr1* expression (Appendix A) (Figure 6B), indicating a persistent, yet altered, relationship between neuroinflammation and cannabinoid receptor signaling during withdrawal.

In the hippocampus, repeated administration of WIN 55,212-2 and HU-210 was associated with significant correlations between gliosis and inflammatory markers (*Iba1, Gfap, Mrc1, Fcgr2b, Ikbkb,* and *Ptgs2*) and components of the endocannabinoid system (*Cnr1, Dagla, Daglb, Napepld,* and *Faah*), suggesting that cannabinoid-induced neuroimmune modulation extends beyond the PFC (Appendix A) (Figure 6C). However, during withdrawal, only a subset of these associations persisted, including correlations between *Gfap, Il1b,* and *Ikbkb* with *Mgll,* as well as *Mrc1, Fcgr2b,* and *Il1b* with *Napepld* and *Faah* (Appendix A) (Figure 6D).

These findings support the hypothesis that synthetic cannabinoid exposure modulates neuroinflammatory responses in a phase-dependent manner, with withdrawal leading to partial resolution of these effects. While correlation analyses do not establish causation, they highlight key molecular relationships that warrant further mechanistic investigation into the immunomodulatory effects of cannabinoid receptor activation.

## 4. Discussion

Our findings revealed that the administration of the synthetic cannabinoids WIN 55,212-2 and HU-210 provoked distinct long-lasting alterations in the expression of the cannabinoid system, modifying the pattern of both the glial response and the inflammatory markers in the PFC and hippocampus (Figure 7). Being part of the corticolimbic system, the PFC is greatly involved in decision-making tasks, including stress-associated responses, whereas the hippocampus is primarily involved in memory formation and learning processes. Alterations in cannabinoid signaling have a differential impact in both regions, which could lead to specific behavioral and cognitive alterations. At the molecular level, these changes might have a direct impact on neuroinflammatory responses, essential for plasticity-based neuroadaptations and potentially influencing synaptic remodeling and cognitive functions [22,23]. Thus, they are relevant to deciphering the effect of two potent synthetic cannabinoids, WIN 55,212-2, a non-selective cannabinoid receptor agonist, and HU-210, a highly potent classical cannabinoid, in cannabinoid signaling.

The scientific literature strongly supports a relevant role for endocannabinoids in modulating the central inflammatory response both in neuronal and glial populations [24,25]. Our results show that both WIN 55,212-2 and HU-210 shared mechanisms of immune modulation during repeated administration and exerted immunosuppression during their withdrawal in the PFC and hippocampus. Interestingly, repeated administration of WIN 55,212-2 was associated with a decrease in the protein levels of the glial marker IBA1 and inflammatory factors NF-κβ and IKKβ. Additionally, CB2, DAGLβ, and NAPE-PLD levels decreased, while FAAH increased. Both WIN 55,212-2 and HU-210 were also associated with decreased CB1 and MAGL levels and increased PPARα levels. If endocannabinoid levels were altered, this could be related to changes in cannabinoid-metabolizing enzymes, which are likely a result of increased exogenous cannabinoid concentrations reaching the PFC and promoting an overactivation that leads to a compensatory decrease in cannabinoid receptor protein levels and desensitization of CB-mediated G protein activation [26,27,28]. Inflammation is usually linked to upregulated expression of CB1 and CB2 [29], and non-psychoactive CB2 activation is seen to decrease microglial-derived neuroinflammatory processes under some [30,31] but not all [32] inflammatory conditions, which underlines the specificity of CB2-mediated immunosuppressive effects. The fact that only WIN 55,212-2, which reduced CB2 protein levels, was able to reduce IBA1, a marker of glial cells, and the inflammatory pathway of NF-κB and its activating molecule IKKβ, suggests that CB2-related mechanisms may play a role in the immunosuppressive effects of WIN 55,212-2. However, given that HU-210, another CB1/CB2 agonist, did not exert the same effect, and that CB2 involvement was not directly tested using an antagonist, alternative mechanisms cannot be ruled out. It is important to note that we did not perform pharmacological blockade of CB1 or CB2 receptors; thus, our findings do not confirm a direct causal relationship between CB2 activation and immunosuppression. In fact, CB2 is expressed in IBA1-positive cells and enhanced in reactive microglia [33], and selective agonists of CB2 are shown to reduce NF-κB activation [34]. The differential effects of WIN 55,212-2 and HU-210 on immunosuppression in the PFC could be related to differences in their receptor binding profiles. HU-210 has a higher affinity and agonistic activity at CB1, which may influence glial activation differently. However, given the complexity of cannabinoid signaling, other receptor-mediated or downstream signaling mechanisms could also be involved.

Opposite to the effects observed in the PFC, WIN 55,212-2 and HU-210 administration caused an increase in the protein levels of glial marker IBA1, accompanied by increased NF-κB, IKKβ, and Vimentin, where only WIN 55,212-2 caused an increase in CB1, and HU-210 was associated with decreased DAGLα. Given the null influence of WIN 55,212-2 and HU-210 on CB2 in the hippocampus, one would expect no overall changes in the glial and inflammatory profile. The intriguing reactivity of astrocytes to repeated WIN 55,212-2 and HU-210 specifically in the hippocampus, but not in the PFC, could be due to regional differences in the functional and morphological response of the PFC and hippocampus to repeatedly exacerbated cannabinoid stimulus. Although WIN 55,212-2 increased CB1 levels in the hippocampus, the presence and expression of CB1 are low or non-existent in the microglia [35], suggesting that this enhanced response of the CB1 is not located in these immune cells and that other possible mechanisms involved in hippocampal inflammatory responses are set in place. This goes in accordance with studies showing that WIN 55,212-2 administration to human astrocytes reduces inflammatory response induced by IL-1β independently of CB1 and PPARα, an effect also observed by WIN 55,212-2 and HU-210 administration to rat microglia [36,37]. It is important to note that most synthetic cannabinoids lack a thorough analysis of target selectivity, so off-target actions have to be considered. This is a relevant issue when exploring new synthetic cannabinoid drugs. Hence, we cannot exclude that further mechanisms could likely contribute to cannabinoid-mediated inflammatory response in the hippocampus during repeated drug administration.

In addition, it is noteworthy that mechanistic in vitro studies and behavioral in vivo experiments usually involve previous activation of glial cells or previous pathologic states to prove the anti-inflammatory response of cannabinoids [38,39,40]. This fact may hinder the basal-state interactions of the cannabinoids with the neuronal and glial systems and elevate the complexity of the immune response to the repeated use of synthetic cannabinoids. Exposure to opioids also produces a higher immunosuppressive response in the PFC compared to the hippocampus, which may imply basal differences between the PFC and the hippocampus to drug cessation [41]. PFC is greatly involved in emotional and stress-related processes and cannabinoid intake exerts profound structural changes in PFC [42,43] that could be related to significant alterations in glial activation and neuroimmune responses as compared to the hippocampus. Since chronic cannabinoid abuse and withdrawal induce emotional-affective behavioral changes, a more robust immunosuppressive response in the PFC may be indicative of a compensatory mechanism to the neuroinflammatory state during cannabinoid abuse.

Drug withdrawal in the PFC was associated with a WIN 55,212-2 specific decrease in IBA1 levels, associated with lower cannabinoid-degrading enzymes FAAH and MAGL. Both WIN 55,212-2 and HU-210 also provoked a marked increase in the protein levels of the AEA-synthetizing enzyme NAPE-PLD. Changes in levels of cannabinoid-metabolizing enzymes are likely to respond to sudden cessation of exogenous cannabinoid supply. Since MAGL primarily hydrolyzes 2-AG and FAAH metabolizes AEA, a decrease in the protein levels of these enzymes is likely to result in an increased availability of endocannabinoids, resulting in a rebound activation of cannabinoid receptors. Withdrawal of HU-210, however, evoked a decrease in protein levels of CB1 and DAGLα, which is responsible for the biosynthesis of 2-AG. This, together with WIN 55,212-2-specific increase in 2-AG synthetizing NAPE-PLD, could be related to an increased accumulation of AEA in favor of 2-AG. This was also true for the hippocampus, where WIN 55-212,2 and HU-210 withdrawal caused decreased protein levels of DAGLα and DAGLβ, possibly indicating a switch favoring AEA signaling. Withdrawal of WIN 55,212-2 and HU-210 also led to a decrease in IBA1, GFAP, and the expression of inflammatory markers *Fcgr2b* and *Rela*, which could be indicative of a resolution phase. Interestingly, AEA has been reported to attenuate age-related microglial activation and decrease the number of IBA-1-positive microglial cells in the hippocampus of rats [44,45]. AEA has low CB2 affinity compared to 2-AG and less impact on CB1 signaling, while it acts also as a ligand for other receptors such as PPARα, which is known to reduce neuroinflammation [46,47]. Thus, other compensatory mechanisms might be involved in drug-cessation reduction in inflammatory markers in the PFC and hippocampus. An interesting candidate is the NAPE-PLD-PPARα axis. NAPE-PLD enzyme is also associated with the production of anti-inflammatory acylethanolamides such as oleoylethanolamide (OEA) that acts through the PPARα receptor, and this axis is altered as the result of the administration of the synthetic cannabinoids used in this study, especially in the prefrontal cortex. The possibility of a net contribution of non-cannabinoid acylethanolamides to the effects observed has a parallel in the well-described effect of OEA as an anti-inflammatory endogenous response to the neuroinflammation associated with alcohol exposure [48]. Altogether, the observed decline in gliosis markers may reflect a resolution phase following the removal of synthetic cannabinoids. While changes in cannabinoid-metabolizing enzymes suggest a potential disruption or downregulation of endocannabinoid signaling, it is important to note that we did not directly measure endocannabinoid levels in this study. Rather, potential changes are inferred based on the observed expression patterns of their metabolic enzymes. Although our findings suggest a potential shift in endocannabinoid signaling, further studies are needed to directly quantify endocannabinoid levels and validate this hypothesis.

Although this study focuses primarily on the molecular alterations of the PFC and hippocampus during repeated drug administration and withdrawal, further studies would benefit from morphological analysis of glial cells to understand the alterations in gliosis and inflammation observed. Moreover, cannabinoid use has been shown to alter the connectivity between the PFC and hippocampus, and thus, specific changes in the aforementioned regions could lead to differential behavioral alterations that should be explored [49]. A limitation of this report is the lack of studies on the female brain. Sexual dimorphism has been observed in the structural alterations caused by cannabinoid abuse, with pronounced volumetric changes in the PFC compared to the hippocampus in female rats, which might be due to the different sensitivity of the dopaminergic system, which is tightly regulated by the cannabinoid system in the PFC [43,50]. Future studies will take into consideration a potential sex-specific sensitivity of the endogenous cannabinoid system to the exogenous administration of synthetic cannabinoids. A study limitation is the selection of WIN55,212-2 and HU-210 as representative synthetic cannabinoids. While WIN55,212-2 is not a widely abused substance, it remains a prototypical aminoalkylindole and a non-selective cannabinoid receptor agonist, acting as a partial agonist at CB1 and a full agonist at CB2 receptors, making it a useful pharmacological tool to assess synthetic cannabinoid-induced neuroinflammatory changes. Similarly, HU-210, though classified as a classical cannabinoid, exhibits significantly greater potency than Δ9-THC and has been found in illicit synthetic cannabinoid mixtures. These differences highlight the need for further studies including a broader range of synthetic cannabinoids to fully capture their diverse effects. Additionally, our study was conducted exclusively in male Wistar rats. There are clear sex-dependent differences in cannabinoid receptor availability in humans and laboratory animals, with female cannabinoid receptors being affected by the estrous cycle [51,52,53]. Because of these hormonal influences and the complexity of studying synthetic cannabinoid responses in females on endocannabinoid system regulation and neuroinflammatory responses, we decided first to characterize these responses in males and in a future study investigate whether similar effects occur in female subjects to provide a more comprehensive understanding of synthetic cannabinoid impact. It is important to note that even vehicle administration can lead to neurobiological changes, as previous studies have shown that repeated handling and injections can induce mild stress responses that affect gene expression in the endocannabinoid system [54,55]. Interestingly, significant alterations in gene and protein expression were also observed in the vehicle group during the withdrawal phase compared to the repeated injection phase. This suggests that repeated exposure to handling and injections may induce lasting neurobiological changes, even in the absence of pharmacological treatment. Additionally, cessation of vehicle administration, which contained ethanol and emulphor, might have contributed to observed differences, as vehicle components can influence lipid homeostasis and receptor regulation. These findings underscore the necessity of considering procedural effects when interpreting drug-induced changes. Such stress-related adaptations may contribute to the observed changes during the withdrawal phase in vehicle-treated animals. Further studies are also needed to determine the extent to which these effects are mediated by stress-induced neuroadaptations versus drug-specific mechanisms.

## 5. Conclusions

These results provide insight into the dynamic alterations of the endocannabinoid signaling and neuroinflammatory responses, which are crucial to understanding the long-term consequences of cannabinoid exposure. Regional differences in the PFC and hippocampus further highlight the glial response’s specific susceptibility and specificity to cannabinoid modulation which could play a role in cognitive and emotional processing, where the PFC is more responsive to synthetic cannabinoid withdrawal, showing a robust suppression of glial activation and a potential compensatory mechanism to cessation of cannabinoid use in contrast to more subtle changes in the hippocampus. Overall, these results highlight the interconnection of the cannabinoid system and the neuroimmune response, and provide a basis for the effects of repeated synthetic cannabinoid use on two brain regions crucial for decision and memory.

## Figures and Tables

**Figure 1 biomolecules-15-00417-f001:**
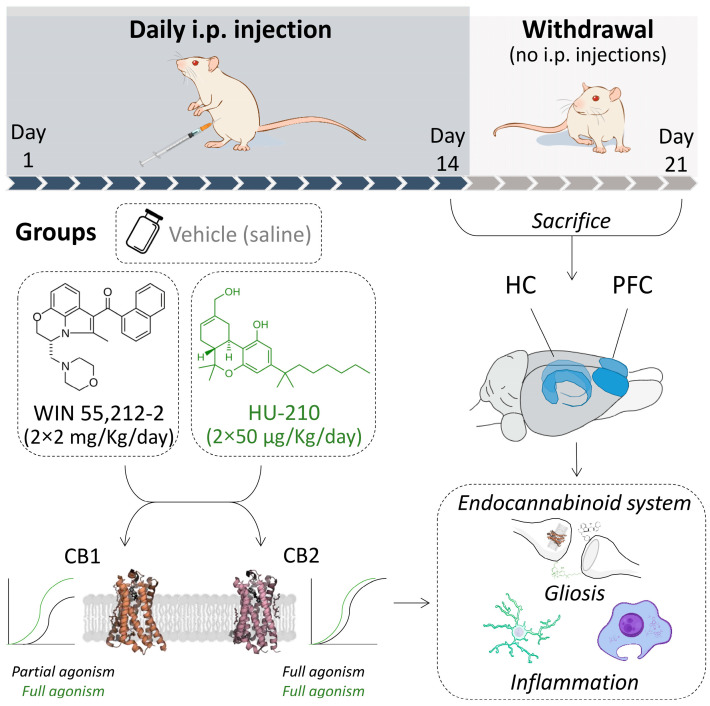
Representative scheme of experimental design involving administration period, doses, brain regions, and systems analyzed. Abbreviations: CB1, cannabinoid receptor type 1; CB2 cannabinoid receptor type 2; HC, hippocampus; PFC, prefrontal cortex. Abbreviations for genes, their protein products, and corresponding functions can be found in Appendix A.

**Figure 2 biomolecules-15-00417-f002:**
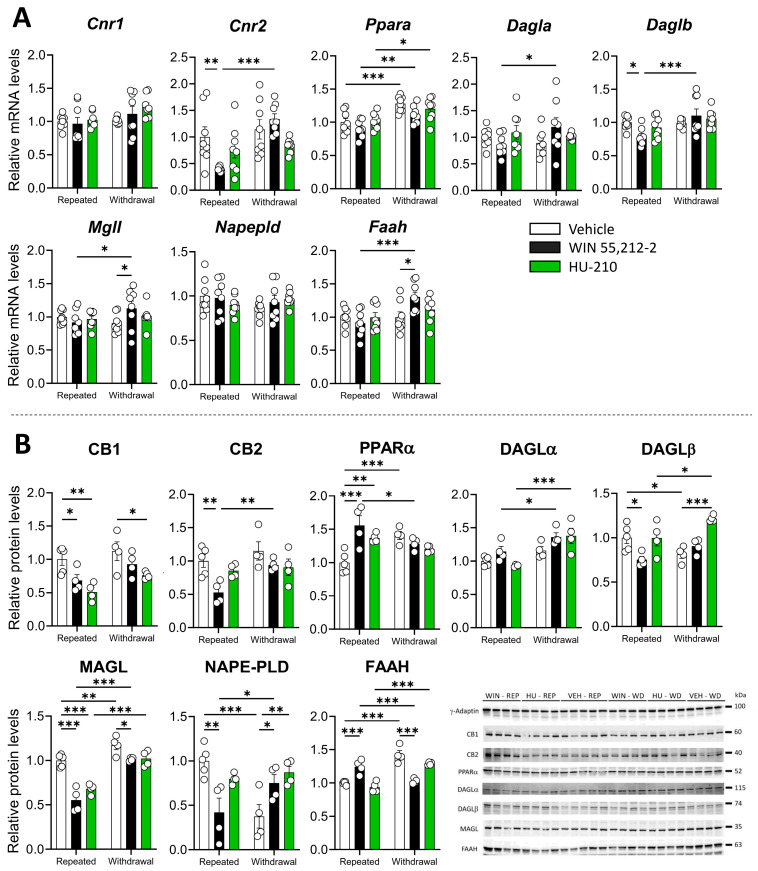
Repeated administration and withdrawal of WIN 55,212-2 and HU-210 on the endocannabinoid system markers of gene expression (**A**) and protein levels (**B**) in the prefrontal cortex. Two-way analysis of variance (ANOVA), whose factors were dependence (repeated administration vs. withdrawal) and drug (vehicle vs. WIN 55,212-2 or HU-210) followed by Tukey post hoc test for multiple comparisons between two groups (*). A *p* value < 0.05 was considered statistically significant (* = *p* < 0.05, ** = *p* < 0.01, *** = *p* < 0.001).

**Figure 3 biomolecules-15-00417-f003:**
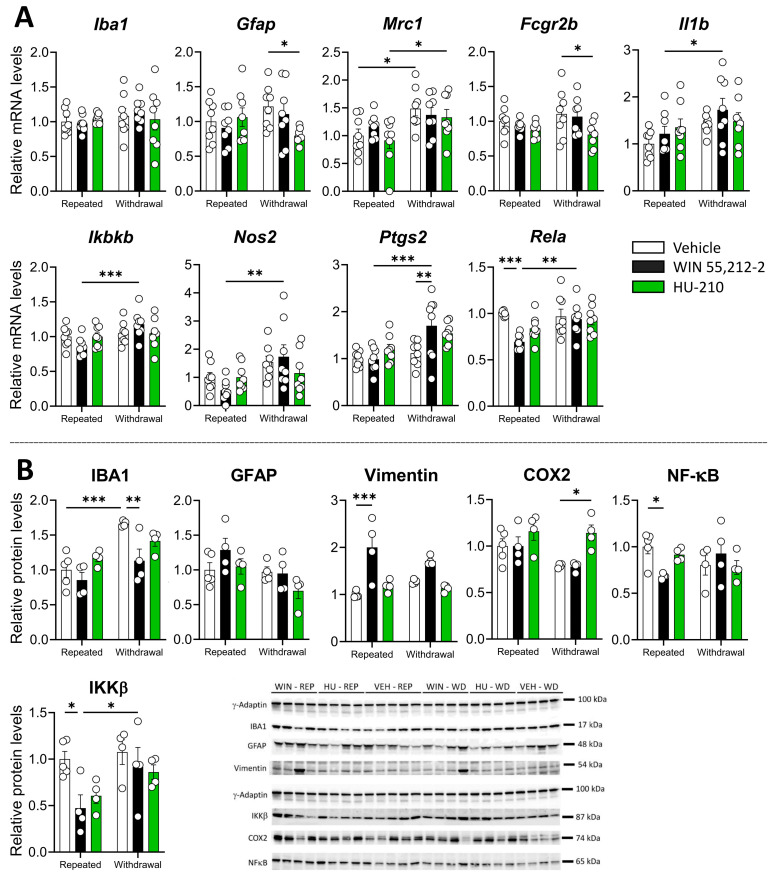
Effects of repeated administration and withdrawal of WIN 55,212-2 and HU-210 on gliosis and inflammatory markers at both the gene expression (**A**) and protein levels (**B**) in the prefrontal cortex. Two-way analysis of variance (ANOVA), whose factors were dependence (repeated administration vs. withdrawal) and drug (vehicle vs. WIN 55,212-2 or HU-210) followed by Tukey post hoc test for multiple comparisons between two groups (*). A *p* value < 0.05 was considered statistically significant (* = *p* < 0.05, ** = *p* < 0.01, *** = *p* < 0.001).

**Figure 4 biomolecules-15-00417-f004:**
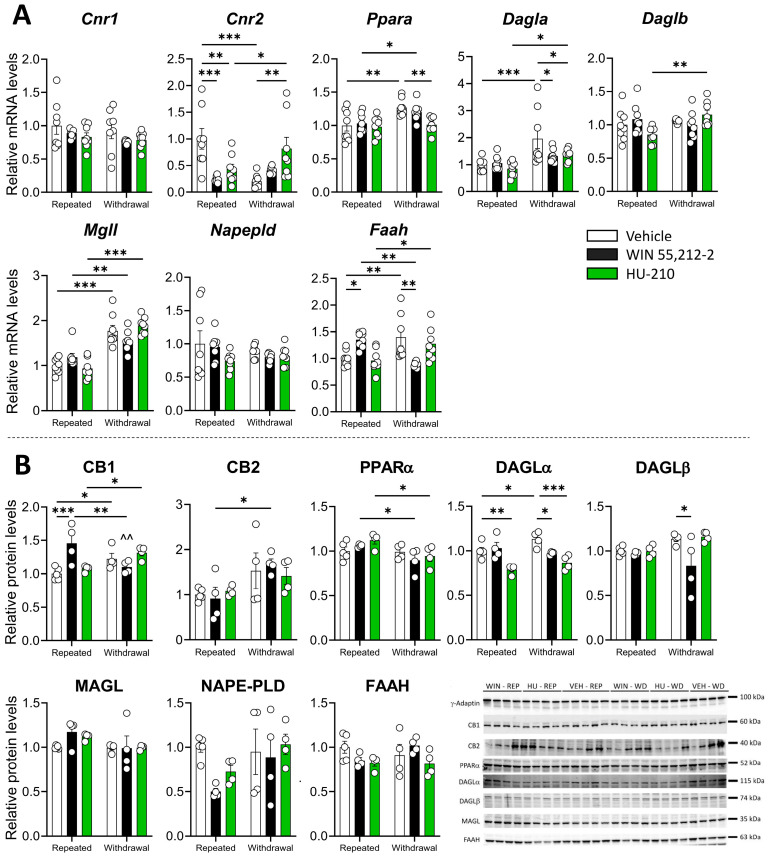
Repeated administration and withdrawal of WIN 55,212-2 and HU-210 on the endocannabinoid system markers of gene expression (**A**) and protein levels (**B**) in the hippocampus. Two-way analysis of variance (ANOVA), whose factors were dependence (repeated administration vs. withdrawal) and drug (vehicle vs. WIN 55,212-2 or HU-210) followed by Tukey post hoc test for multiple comparisons between two groups (*). A *p* value < 0.05 was considered statistically significant (* = *p* < 0.05, ** = *p* < 0.01, *** = *p* < 0.001).

**Figure 5 biomolecules-15-00417-f005:**
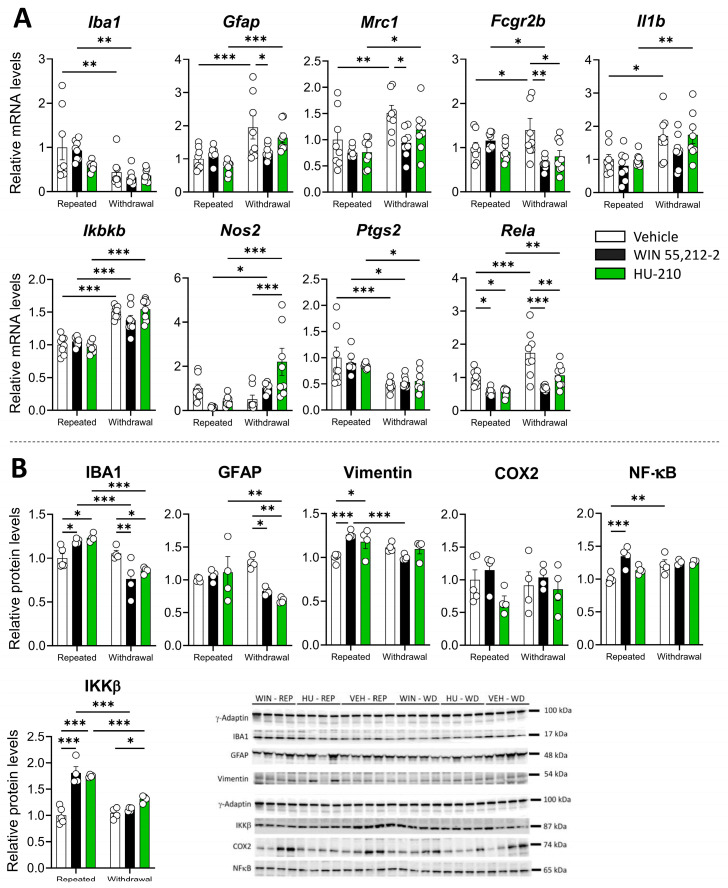
Repeated administration and withdrawal of WIN 55,212-2 and HU-210 on gliosis and inflammatory markers of gene expression (**A**) and protein levels (**B**) in the hippocampus. Two-way analysis of variance (ANOVA), whose factors were dependence (repeated administration vs. withdrawal) and drug (vehicle vs. WIN 55,212-2 or HU-210) followed by Tukey post hoc test for multiple comparisons between two groups (*). A *p* value < 0.05 was considered statistically significant (* = *p* < 0.05, ** = *p* < 0.01, *** = *p* < 0.001).

**Figure 6 biomolecules-15-00417-f006:**
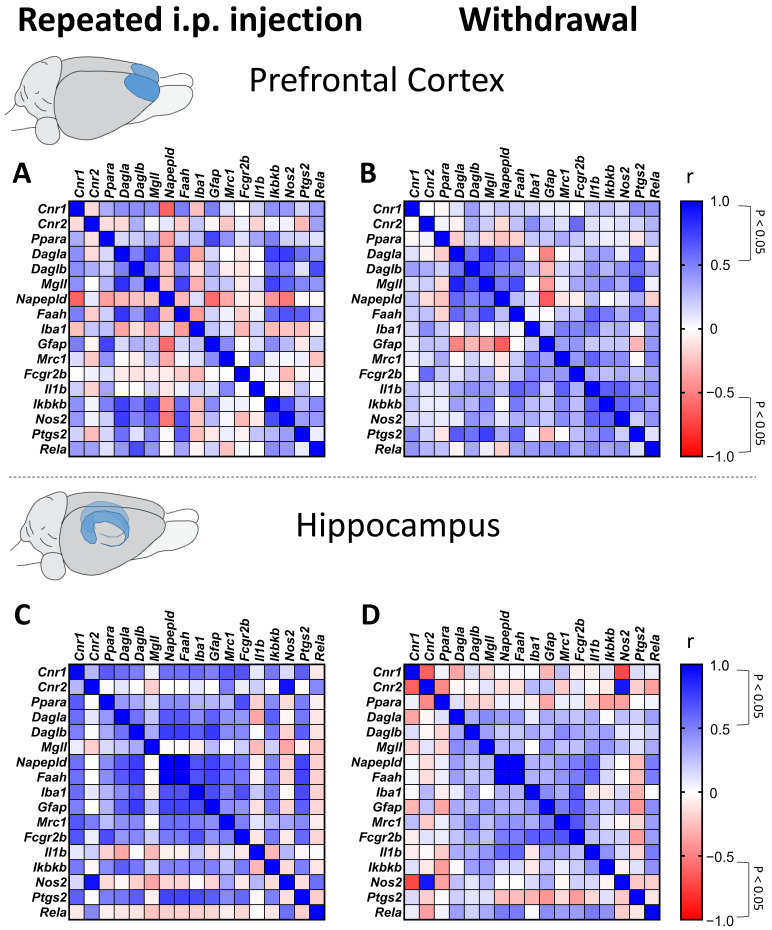
Spearman correlation analysis of gene expression of endocannabinoid and gliosis/inflammation markers during repeated administration (**A**) and withdrawal in the prefrontal cortex (**B**) and repeated administration (**C**) and withdrawal (**D**) in the hippocampus. A *p* level of < 0.05 was considered significant.

**Figure 7 biomolecules-15-00417-f007:**
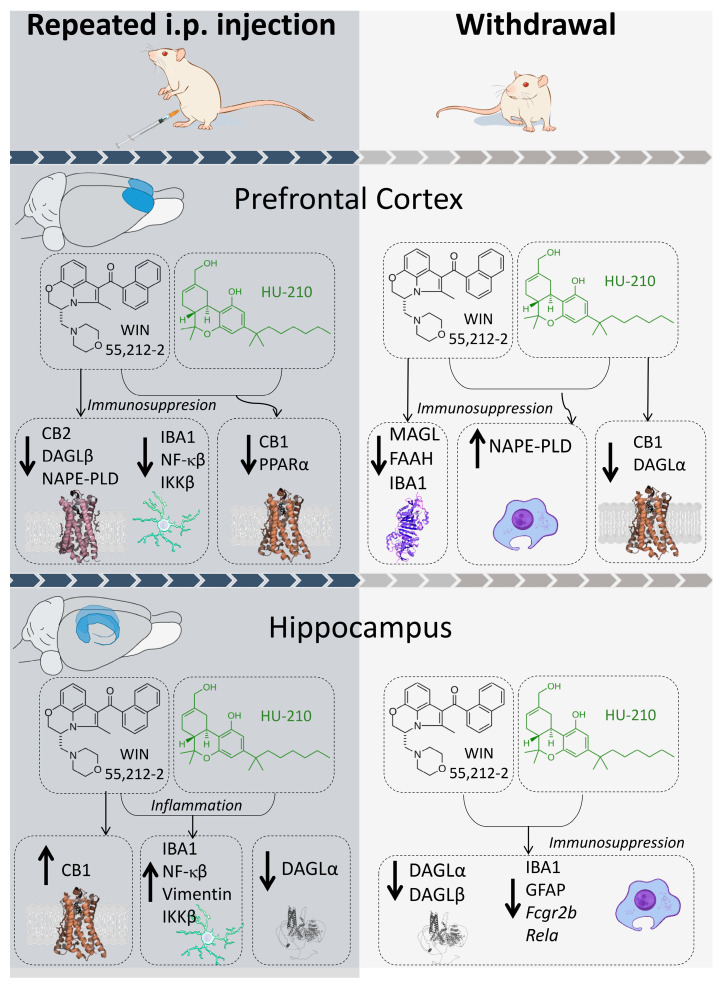
Graphical representation of main results observed for repeated administration and withdrawal of WIN 55,212-2 and HU-210 on endocannabinoid and gliosis/inflammation markers in the prefrontal cortex and hippocampus.

## Data Availability

Data presented in this study are available upon request from the corresponding author.

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
