# Peer review of "Region-Specific Impact of Repeated Synthetic Cannabinoid Exposure and Withdrawal on Endocannabinoid Signaling, Gliosis, and Inflammatory Markers in the Prefrontal Cortex and Hippocampus"

_biomolecules, 2025, doi:10.3390/biom15030417_

Round 1

Reviewer 1 Report

Comments and Suggestions for Authors

Vadas et al. have treated rats with the cannabinoid receptor agonists WIN55,212-2 or HU-210 twice daily. Part of the animals was killed after 14 days whereas another group was treated with vehicle for another 7 days before biochemical parameters were studied in the prefrontal cortex and hippocampus. Gene expression and protein of components of the endocannabinoid system, of glial markers and of inflammatory parameters were quantified. 

The study appears to be sound and the results are presented in a clear manner. I have very few points only:

  • l. 3, 207 and 289 and other sites of the manuscript: It is not really clear what the authors mean when they say "differentially affects": WIN vs. HU; prefrontal cortex vs. hippocampus; repeated administration vs. withdrawal; glial vs. inflammatory parameters?
  • l. 41: If one reads some reviews about the medical use of cannabinoids, one gets the impression that cannabinoids are good for almost every disorder. As a physician, I agree with the indications pain and nausea and I would suggest to add anorexia, some types of epilepsies and spasticity associated with multiple sclerosis instead of cerebral trauma etc.
  • l. 52-57: This is a very long sentence I did not fully understand. It should be divided into shorter parts.
  • l. 141: Authors should write "2 x 2 mg/Kg per day" and "2 x 50 µg/Kg per day" for WIN and HU, respectively. Moreover, the abbreviations used in the figure should be explained. An even better suggestion would be that you add a table in which the abbreviations for the gene and the protein and the function of each entity are given, e.g. cnr1 (gene), CB1 (protein) and cannabinoid receptor type 1, respectively.
  • l. 224: I do not understand why the mRNA expression or protein quantity is affected in the withdrawal experiments in animals which have consistently exposed to vehicle.
  • l. 234: I do also not understand why WIN had an inhibitory effect on e.g. CB1 and CB2 protein whereas HU inhibited CB2 only. I would have expected that CB2 protein is inhibited by either agonist whereas CB1 is affected by HU only (since WIN is a partial agonist at this receptor).
  • l. 375: The "in" between "to" and "the" appears to be superfluous.
  • l. 402: This sentence appears to be incomplete.
  • l. 448: "an" instead of "and" appearing in the first half of the line?

Author Response

Point-by-point Response to Reviewer 1 Comments

Thank you for your time and effort in reviewing our manuscript. Below, we provide detailed responses to each of your comments. The corresponding revisions and corrections have been highlighted in yellow in the re-submitted files.

Comment 1.1:

l. 3, 207 and 289 and other sites of the manuscript: It is not really clear what the authors mean when they say "differentially affects": WIN vs. HU; prefrontal cortex vs. hippocampus; repeated administration vs. withdrawal; glial vs. inflammatory parameters?

Response 1.1:

Thank you for raising this important question. We acknowledge the need for further clarification regarding the phrase "differentially affect." In our study, this term refers to the distinct effects observed between (1) WIN 55,212-2 and HU-210, (2) different brain regions (prefrontal cortex vs. hippocampus), (3) different experimental conditions (repeated administration vs. withdrawal), and (4) different biological parameters (glial vs. inflammatory markers).

To improve clarity, we have revised these sections as follows:

-        Title (line 3): Region-Specific Impact of Repeated Synthetic Cannabinoid Exposure and Withdrawal on Endocannabinoid Signaling, Gliosis, and Inflammatory markers in the Prefrontal Cortex and Hippocampus

-        Line 207: Reworded "differentially modify" to "lead to distinct alterations", to explicitly state that the effects of repeated administration and withdrawal differ in specific ways.

Updated text: “Repeated administration and withdrawal of WIN 55,212-2 and HU-210 lead to distinct effects on the endocannabinoid system in the prefrontal cortex” on page 6, lines 216-217

-        Line 289: Adjusted the phrasing to improve readability and clarify the comparison.

Updated text:Effects of repeated administration and withdrawal of WIN 55,212-2 and HU-210 on gliosis and inflammatory markers at both the gene expression (A) and protein (B) levels in the prefrontal cortex.” on page 8, lines 291-293.

We hope these modifications address your concern by making the descriptions more explicit. Please let us know if further refinements are needed.

Comment 1.2:
l. 41: If one reads some reviews about the medical use of cannabinoids, one gets the impression that cannabinoids are good for almost every disorder. As a physician, I agree with the indications pain and nausea and I would suggest to add anorexia, some types of epilepsies and spasticity associated with multiple sclerosis instead of cerebral trauma etc.

Response 1.2:

Thank you for your suggestion. We have revised the text accordingly on page 2, lines 43-47.

Updated text:

“Cannabis and synthetic cannabinoids are increasingly used recreationally, as well as medically to treat pain, nausea, depressive symptoms, anorexia related to weight loss, tumors, tremors, and spasticity associated with multiple sclerosis, along with other neurological disorders such as epilepsy, Alzheimer’s, Parkinson’s and Huntington’s disease [1].”

Comment 1.3:

l. 52-57: This is a very long sentence I did not fully understand. It should be divided into shorter parts.

Response 1.3:

Thank you for pointing this out. We agree with this comment. Therefore, we have revised the section accordingly. The changes can be found on page 2, lines 56-62.

Updated text:
"As an example, chronic administration of natural cannabinoids leads to tolerance of acute effects such as hypoactivity, lethargy, and hypothermia
. Consequently, cessation of cannabinoid use can trigger a withdrawal syndrome that includes anxiety and tachycardia. Consumption and withdrawal from synthetic cannabinoids, however, may result in a severe neurological syndrome, often accompanied by psychotic symptoms and, in some cases, even death [5-7]."

Comment 1.4:

l. 141: Authors should write "2 x 2 mg/Kg per day" and "2 x 50 µg/Kg per day" for WIN and HU, respectively. Moreover, the abbreviations used in the figure should be explained. An even better suggestion would be that you add a table in which the abbreviations for the gene and the protein and the function of each entity are given, e.g. cnr1 (gene), CB1 (protein) and cannabinoid receptor type 1, respectively.

Response 1.4:

Thank you for your comment. The requested change regarding the dosing format has been implemented in text and Figure 1, where the information is now presented as "2 × 2 mg/kg/day" and "2 × 50 µg/kg/day" for WIN 55,212-2 and HU-210, respectively.

In addition, we appreciate your valuable suggestion regarding the abbreviations. In response, we have created a table detailing the gene names, their corresponding proteins, and their functions. This table has been added as Supplementary Table 1 (Table S1).

Comment 1.5:

l. 224: I do not understand why the mRNA expression or protein quantity is affected in the withdrawal experiments in animals which have consistently exposed to vehicle.

Response 1.5:

Thank you for your insightful comment. The observed changes in mRNA expression in the vehicle-treated animals during withdrawal may be attributed to factors beyond direct drug effects. Specifically, the vehicle administration itself, coupled with the experimental conditions and withdrawal paradigm, may induce neuroadaptations in the endocannabinoid system. Prior studies have shown that repeated handling, injections, and the withdrawal process can lead to stress-induced alterations in gene expression, even in vehicle-treated animals. These effects are well-documented in the literature, suggesting that the withdrawal phase is not solely reflective of drug-specific adaptations but may also involve broader neurobiological responses. To clarify this point, we have now added a brief explanation in the discussion section of the manuscript. See lines 556-559, and 567-571.
Updated text:

"It is important to note that even vehicle administration can lead to neurobiological changes, as previous studies have shown that repeated handling and injections can induce mild stress responses that affect gene expression in the endocannabinoid system. [54,55]
Such stress-related adaptations may contribute to the observed changes during the withdrawal phase in vehicle-treated animals. Further studies are needed to determine the extent to which these effects are mediated by stress-induced neuroadaptations versus drug-specific mechanisms." 

References:
54.       Gray, J.D.; Rubin, T.G.; Hunter, R.G.; McEwen, B.S. Hippocampal gene expression changes underlying stress sensitization and recovery. Molecular psychiatry 2014, 19, 1171-1178.

55.       Sun, Z.; Cai, D.; Yang, X.; Shang, Y.; Li, X.; Jia, Y.; Yin, C.; Zou, H.; Xu, Y.; Sun, Q. Stress response simulated by continuous injection of ACTH attenuates lipopolysaccharide-induced inflammation in porcine adrenal gland. Frontiers in Veterinary Science 2020, 7, 315.

Comment 1.6:

l. 234: I do also not understand why WIN had an inhibitory effect on e.g. CB1 and CB2 protein whereas HU inhibited CB2 only. I would have expected that CB2 protein is inhibited by either agonist whereas CB1 is affected by HU only (since WIN is a partial agonist at this receptor).

Response 1.6:

Reviewer is raising an interesting point about receptor regulation, ligand-induced changes in protein expression, and the potential mechanisms at play, that is, the differences in the action of WIN55,212-2 and HU-210 on the CB1 and CB2 receptor expression. CB1 and CB2 receptors are both part of the endocannabinoid system, but they are located in different tissues and play different roles. CB1 is primarily found in the central nervous system (CNS), whereas CB2 is more prevalent in peripheral tissues, including immune cells. In this context, receptor downregulation is a common phenomenon, where continuous or chronic stimulation of a receptor by an agonist leads to a reduction in the number of functional receptors on the cell surface through feedback mechanisms such as receptor desensitization, internalization, transcriptional silencing and/or degradation. This is typically a homeostatic response to prevent overstimulation of the cell and excessive signaling through a receptor, and maintain balance. Focusing on your comment:

HU-210 is considered a full agonist at both CB1 and CB2 receptors. Over time, stronger and more sustained activation of receptors by HU-210 can lead to receptor downregulation, meaning fewer receptors are expressed. This effect is particularly notable for CB2 in the hippocampus, where repeated HU-210 may cause a great mRNA downregulation at this receptor similar to WIN55,212-2. At CB1, this effect can be observed in the prefrontal cortex at protein levels, causing a greater downregulation at this receptor compared to WIN55,212-2.

On the other hand, WIN55,212-2 is considered a dual agonist of both CB1 and CB2 receptors, although not as effective as a full agonist like HU-210. This means it can bind to and activate both CB1 and CB2, but with lower efficacies at each receptor. Specifically, WIN55,212-2 is often described as a partial agonist at CB1, meaning it does not fully activate the receptor, does so less robustly, but still exerts a functional effect. This could mean that the putative downregulation of CB1 is less pronounced than what we would expect from a full agonist like HU-210. In fact, repeated WIN55,212-2 did not reduce CB1 protein expression in the prefrontal cortex, probably because its partial agonistic effect is not sufficient to initiate a feedback loop. Moreover, repeated WIN55,212-2 increased CB1 protein expression in the hippocampus, which might be as a result of a feedback loop of increased receptor availability. At CB2, it tends to have a stronger agonistic effect. This could mean that the sustained activation of CB2 by WIN55,212-2 could still reduce CB2 expression due to its strongest agonistic effect, which might lead to a feedback loop of decreased receptor availability. This effect is particularly notable for mRNA CB2 levels in the prefrontal cortex and hippocampus. In addition, repeated WIN55,212-2 may cause a greater downregulation at protein levels of CB2 in the prefrontal cortex compared to HU-210.

WIN55,212-2 and HU-210 would be also expected to have different effects on receptor expression depending on the tissue type they are acting on. More robust effects on downregulation of receptors occur especially in brain tissues where the receptors are expressed at higher levels. CB1 is more concentrated in both glutamatergic and GABAergic neurons of the hippocampus, while CB1 receptors are widely distributed in the layer I and V of the prefrontal cortex, including dopaminergic, glutamatergic and GABAergic neurons. CB2 receptor, with lower presence in brain, is primary expressed in neuroimmune cells under inflammatory conditions.

In summary, both ligand-induced feedback mechanisms and receptor localization in brain may contribute to the differential effects of the action of repeated stimulation of WIN55,212-2 and HU-210 on the CB1 and CB2 receptor expression.

Comment 1.7:

l. 375: The "in" between "to" and "the" appears to be superfluous.

Response 1.7: Thank you for your comment. The sentence in question has been removed, as we have completely revised the paragraph in response to another comment from Reviewer 2. As a result, the original issue no longer appears in the manuscript.

Comment 1.8:
l. 402: This sentence appears to be incomplete.

Response 1.8:

Thank you for your valuable feedback. We acknowledge that the sentence was incomplete and have revised it for clarity.

The revision provides a clearer connection between neuroinflammatory responses and their role in synaptic plasticity and cognitive processes, as supported by the referenced studies.
The updated sentence now reads (on page 13, lines 414-417):

Updated text:

"At the molecular level, these changes might have a direct impact on neuroinflammatory responses, which are essential for plasticity-based neuroadaptations, potentially influencing synaptic remodeling and cognitive functions [22,23]."

Comment 1.9:

l. 448: "an" instead of "and" appearing in the first half of the line?

Response 1.9:

Thank you for your careful review. We have checked the sentence to make the necessary correction. Line 470.

Reviewer 2 Report

Comments and Suggestions for Authors

This manuscript describes observations that two synthetic cannabinoid receptor ligands - WIN-55,212-2 ("WIN") and HU-210 ("HU") - differentially alter mRNA and protein expression levels of various proteins in the endocannabinoid and gliosis/inflammatory systems. Two brain regions were studied - the PFC and hippocampus - and expression levels were determined either during the drug taking phase or 7 days thereafter (termed the "withdrawal" phase). 

Several issues need addressing before publication (in order of appearance in the text). 

p.3: The implied purpose of the study is to determine how synthetic cannabinoids in clandestine use impact physiology; however, despite being an aminoalkylindole, WIN is not readily abused in any country, and the use of WIN as a representative aminoalkylindole is suspect. Same issue with HU as possibly representative of classical cannabinoids. Further, only male Wistar rats were used: are these results translatable to females? These limitations need to be elaborated in the discussion. 

line 125: WIN is not "a selective, high-affinity CB2 agonist." Elsewhere, it is stated that WIN is a CB1 partial agonist; the literature is not entirely sold on this definition. It might be best to compare the efficacy ranges between compounds, rather than label them as having substantially different pharmacodynamic profiles.

2.4 Experimental Design (134): The rationale for why 7 days post-repeated injection was chosen to determine "withdrawal" needs to be explained. What withdrawal signs were monitored, and how did those change over time? Are we past the peak withdrawal, are they still in withdrawal, etc. 

3. Results (206, and throughout the manuscript). The term "addiction" or "addiction effect" shows up a lot, yet "addiction" is not measured in this study. Because the authors designate a "withdrawal" phase, they likely mean "dependence" instead. As above, we do not know if they are truly dependent unless we are monitoring signs of withdrawal. Throughout these results, the term "addiction" should be replaced by the proper terms. If withdrawal was not explicitly monitored, this term should be replaced, too. "Injection phase" and "post-injection phase" might be most accurate. 

In several figures, there are places where gene and/or protein expression levels are significantly altered during the "withdrawal" phase for vehicle when compared to the "repeated injection" phase. Why is that? Why this is and how this influences interpretation of the results needs to be addressed. 

line 258: the section title is misleading because some markers went up in either the repeat administration or post-administration phases, in contrast to the title that suggests that both drugs "decrease markers of gliosis and inflammatory factors." This simplification also appears later (line 277-278) where it is stated definitively that there is an "immunosuppressive response" that isn't shown (were the immune systems of the rats actually challenged at any time?). 

3.5 (starting line 361): It is not clear that there is any hypothesis being tested in this section; this is dredging through data in search of finding correlations. This section, and the associated figure and SI tables, add nothing to the story and need to be removed. 

line 417 (and maybe elsewhere; lots of this in the paragraph starting line 469): it is definitively stated that there are "decreased endocannabinoid levels," which appears to be based on the observation that the expression levels of certain catabolic enzymes were altered; however, it does not appear that endocannabinoid levels were, indeed, determined. In the absence of this confirmation, one cannot make this definitive statement. At best, this language can be softened to say, "if endocannabinoid levels were altered..." and to include a clarifying statement that such levels were not, in fact, determined. 

Same comments in the discussion (469). Lots of commentary speculating what might happen if endocannabinoid levels were altered, but no confirmation that they were. In the absence of this confirmation, all of this is not very convincing of a mechanism. 

lines 426-433: there is a lot of unfounded speculation into mechanisms in this section. For example, why would CB2 agonism be behind WIN's immunosuppressive effects when 1) that was not tested using a CB2 antagonist, 2) HU - which is also a CB1/CB2 agonist - did not cause the same effect, and 3) there are potentially many other mechanisms (line 449)? These speculative comments need to be reexamined in light of the fact that these studies did not attempt to connect CB1 or CB2 activity to the observed effects. 

SI: the two-way ANOVA interaction tables (that include interaction, addiction, drug) are confusing. It is not clear what we are looking at. 

Author Response

Point-by-point Response to Reviewer 2 Comments

Thank you for your time and effort in reviewing our manuscript. Below, we provide detailed responses to each of your comments. The corresponding revisions and corrections have been highlighted in yellow in the re-submitted files.

Comment 2.1:

p.3: The implied purpose of the study is to determine how synthetic cannabinoids in clandestine use impact physiology; however, despite being an aminoalkylindole, WIN is not readily abused in any country, and the use of WIN as a representative aminoalkylindole is suspect. Same issue with HU as possibly representative of classical cannabinoids. Further, only male Wistar rats were used: are these results translatable to females? These limitations need to be elaborated in the discussion.

Response 2.1:

We appreciate your valuable feedback and acknowledge the points raised regarding the selection of WIN55,212-2 and HU-210. While WIN55,212-2 is not commonly associated with illicit drug use, it serves as a well-characterized aminoalkylindole and a non-selective cannabinoid receptor agonist, acting as a partial agonist at CB1 and a full agonist at CB2 receptors. Similarly, HU-210, despite being a classical cannabinoid, is significantly more potent than Δ9-THC and has been detected in illicit cannabinoid preparations. The rationale for their inclusion in our study was to examine the differential effects of synthetic cannabinoids with distinct chemical structures and receptor affinities on the endocannabinoid system and neuroinflammatory processes. We will further elaborate on these considerations in the Discussion section.

Regarding the use of only male Wistar rats, we recognize the importance of sex differences in cannabinoid pharmacology and neuroinflammatory responses. While our study focused on male subjects to minimize variability, future research should investigate potential sex-dependent effects. We will revise the Discussion to address this limitation more explicitly.

Updated text in Discussion:

“A study limitation is the selection of WIN55,212-2 and HU-210 as representative synthetic cannabinoids. While WIN55,212-2 is not a widely abused substance, it remains a prototypical aminoalkylindole and a non-selective cannabinoid receptor agonist, acting as a partial agonist at CB1 and a full agonist at CB2 receptors, making it a useful pharmacological tool to assess synthetic cannabinoid-induced neuroinflammatory changes. Similarly, HU-210, though classified as a classical cannabinoid, exhibits significantly greater potency than Δ9-THC and has been found in illicit synthetic cannabinoid mixtures. These differences highlight the need for further studies including a broader range of synthetic cannabinoids to fully capture their diverse effects.

Additionally, our study was conducted exclusively in male Wistar rats. There are clear sex-dependent differences on cannabinoid receptors availability in humans and laboratory animals, with female cannabinoid receptors being affected by the estrous cycle [51,52,53]. Because of these hormonal influences, the complexity of studying synthetic cannabinoid responses in females on endocannabinoid system regulation and neuroinflammatory responses, we decided first to characterize these responses in males and in a future study investigate whether similar effects occur in female subjects to provide a more comprehensive understanding of synthetic cannabinoid impact."

We have added this limitation to the manuscript (lines 540-556) to fully acknowledge the need of pharmacological and toxicological studies in females.

References:
51. Rodríguez de Fonseca, F.; Cebeira, M.; Ramos, J.A.; Martín, M.; Fernández-Ruiz, J.J. Cannabinoid receptors in rat brain areas: sexual differences, fluctuations during estrous cycle and changes after gonadectomy and sex steroid replacement. Life Sciences. 1994, 54, 159-170.

52. Farquhar, C.E.; Breivogel, C.S.; Gamage, T.F.; Gay, E.A.; Thomas, B.F.; Craft, R.M.; Wiley, J.L. Sex, THC, and hormones: Effects on density and sensitivity of CB1cannabinoid receptors in rats. Drug and Alcohol Dependence. 2019, 194, 20-27.

Comment 2.2:

line 125: WIN is not "a selective, high-affinity CB2 agonist." Elsewhere, it is stated that WIN is a CB1 partial agonist; the literature is not entirely sold on this definition. It might be best to compare the efficacy ranges between compounds, rather than label them as having substantially different pharmacodynamic profiles.

Response 2.2:

Thank you for your insightful comment regarding the pharmacological characterization of WIN 55,212-2. We acknowledge that the description of WIN 55,212-2 as a “selective, high-affinity CB2 agonist” was not precise. To address this, we have revised the manuscript to clarify that WIN 55,212-2 is a *non-selective cannabinoid receptor agonist, acting as a partial agonist at CB1 and a full agonist at CB2.

Additionally, we recognize the ongoing discussion in the literature regarding the exact pharmacodynamic profile of WIN 55,212-2. While it is often classified as a partial CB1 agonist, its efficacy varies depending on the experimental conditions and receptor expression levels. As suggested, we have modified the text to focus on comparing the efficacy differences between WIN 55,212-2 and HU-210 rather than making categorical distinctions in their pharmacological profiles.

These revisions can be found in the Introduction, in the relevant discussion of their effects on neurotransmission, in the Methods section and in the Discussion. We appreciate your suggestion, which has helped improve the accuracy and clarity of our manuscript. 

Updated texts in Introduction:

“WIN 55,212-2 is a non-selective cannabinoid receptor agonist with partial agonist activity at CB1 receptor and full agonist activity at CB2 receptor. In acute administration, it modulates neuronal activity by inhibiting GABAergic and glutamatergic synaptic neurotransmission [14]. Repeated administration of WIN 55,212-2 induces long-lasting effects during withdrawal in the dopaminergic system associated with the psychoactive rewarding and locomotor effects [15]. HU-210 is a highly potent full agonist at both CB1 and CB2 receptors, with extended half-life. It exerts a stronger inhibitory effect on GABAergic neurotransmission in the hippocampus compared to WIN 55.212-2 and THC [16].” on page 2, lines 86-93

Updated texts in Materials and Methods:

“WIN 55,212-2 ((R)-(+)-[2,3-Dihydro-5-methyl-3-(4-morpholinylmethyl)pyrrolo[1,2,3-de]-1,4-benzoxazin-6-yl]-1-naphthalenylmethanone mesylate; cat. no. 1038, Tocris, Bristol, UK), a cannabinoid receptor agonist with high affinity for CB2 (Ki = 3.3 nM) and partial agonist activity at CB1, was dissolved in a vehicle composed of saline with 5% of ethanol and 5% of emlphor and made fresh every day.” on page 3, line 129-133

Updated texts in Discussion:

“Thus, the relevance to deciphering the effect of WIN 55,212-2, a non-selective cannabinoid receptor agonist, and HU-210, a highly potent classical cannabinoid, in the cannabinoid signaling.” on page 13, lines 417-419

“The fact that only WIN 55,212-2, which reduced CB2 protein levels, was able to reduce IBA1, a marker of glial cells, and the inflammatory pathway of NF-κB and its activating molecule IKKβ, suggests that CB2-related mechanisms may play a role in the immunosuppressive effects of WIN 55,212-2. page 14, lines 440-444

“The differential effects of WIN 55,212-2 and HU-210 on immunosuppression in the PFC could be related to differences in their receptor binding profiles. HU-210 has a higher affinity and agonistic activity at CB1, which may influence glial activation differently.” page 14, lines 450-453

Comment 2.3:

2.4 Experimental Design (134): The rationale for why 7 days post-repeated injection was chosen to determine "withdrawal" needs to be explained. What withdrawal signs were monitored, and how did those change over time? Are we past the peak withdrawal, are they still in withdrawal, etc.

Response 2.3:

Thank you for your insightful question. We recognize the importance of defining the withdrawal timeline in the context of synthetic cannabinoid exposure. This is a very relevant question due to the scarcity of detailed pharmacokinetic data for these synthetic cannabinoids, as well as for their metabolic and elimination pathways. Thus, the only approach to establish the withdrawal peak is based in our experience with these drugs and the reports published on dependence and withdrawal.

In this study, we selected the 7-day post-administration time point to assess longer-lasting neuroadaptations in the endocannabinoid and neuroimmune systems rather than acute withdrawal symptoms. Previous research indicates that cannabinoid withdrawal symptoms—such as irritability, hyperalgesia, and alterations in locomotor activity—typically peak within the first 24–72 hours after drug cessation and gradually diminish thereafter. While the exact duration of withdrawal symptoms may vary, studies suggest that the most pronounced behavioral signs tend to subside within the first several days post-administration. [1,2,3]

Although we did not explicitly assess withdrawal symptoms in this study, prior research suggests that by 7 days post-administration, animals are likely beyond the peak withdrawal phase, and overt behavioral withdrawal signs are expected to be reduced. However, neurochemical and synaptic adaptations may persist beyond this period, even in the absence of obvious behavioral withdrawal symptoms. In the case of HU-210, previous work from our lab indicates that at 4 days after treatment cessation, animals have recovered from sedation imposed by this potent synthetic cannabinoid, but neuradaptions in the sensitivity to dopamine receptors agonists are present. Notably, previous research has demonstrated that CB1 receptor downregulation and desensitization remain evident at this time point following chronic cannabinoid exposure [4]. This timeframe was chosen to capture sustained molecular alterations in the prefrontal cortex and hippocampus, which are relevant to cannabinoid dependence, neuroplasticity, and neuroimmune regulation.

We appreciate the reviewer’s thoughtful question and hope this clarification provides further insight into our rationale for selecting this withdrawal period.

1. Varvel, S.A.; Bridgen, D.T.; Tao, Q.; Thomas, B.F.; Martin, B.R.; Lichtman, A.H. Δ9-Tetrahydrocannbinol Accounts for the Antinociceptive, Hypothermic, and Cataleptic Effects of Marijuana in Mice. Journal of Pharmacology and Experimental Therapeutics 2005, 314(1), 329-37.

2. Schlienz, N.J.; Budney, A.J.; Lee, D.C.; Vandrey, R. Cannabis Withdrawal: a Review of Neurobiological Mechanisms and Sex Differences. Current Addiction Reports 2017, 4, 75–81.

3. Moreno M, Lopez-Moreno JA, Rodríguez de Fonseca F, Navarro M. Behavioural effects of quinpirole following withdrawal of chronic treatment with the CB1 agonist, HU-210, in rats. Behavioural Pharmacology. 2005, 16, 441-446.

4. Rodríguez De Fonseca, F.; Gorriti, M.; Fernandez-Ruiz, J.; Palomo, T.; Ramos, J. Downregulation of rat brain cannabinoid binding sites after chronic Δ9-tetrahydrocannabinol treatment. Pharmacology Biochemistry and Behavior 1994, 47, 33-40.

Comment 2.4:

3. Results (206, and throughout the manuscript). The term "addiction" or "addiction effect" shows up a lot, yet "addiction" is not measured in this study. Because the authors designate a "withdrawal" phase, they likely mean "dependence" instead. As above, we do not know if they are truly dependent unless we are monitoring signs of withdrawal. Throughout these results, the term "addiction" should be replaced by the proper terms. If withdrawal was not explicitly monitored, this term should be replaced, too. "Injection phase" and "post-injection phase" might be most accurate.

Response 2.4:

We appreciate the comments of the reviewer. It is true that we are not truly addressing addiction (a complex behavior) but particular aspects of addiction (tolerance/dependence and withdrawal). Our protocol of administration is based in previous studies in rats and mice, where we confirmed the induction of tolerance and the presence of dependence by revealing an antagonist-precipitated withdrawal [1,2]. Thus, the correct terms should be dependence and withdrawal. We have changed the terms throughout the manuscript. Following referee’s report.

Comment 2.5:

In several figures, there are places where gene and/or protein expression levels are significantly altered during the "withdrawal" phase for vehicle when compared to the "repeated injection" phase. Why is that? Why this is and how this influences interpretation of the results needs to be addressed.

Response 2.5:

We appreciate the reviewer’s observation regarding significant changes in gene and/or protein expression levels in the vehicle group during the withdrawal phase. These differences could stem from multiple factors:

The repeated handling and injections over 14 days likely induced acute stress responses in the vehicle group, as evidenced by behavioral changes observed in previous studies [1]. While the long-term neurobiological adaptations remain unclear, prior research suggests that repeated exposure to experimental procedures, even in control conditions, can influence stress-related behaviors and physiological states. These effects could contribute to the observed alterations during the withdrawal phase.

The cessation of vehicle injections may itself induce adaptive responses, as repeated administration of a vehicle solution containing ethanol and emulphor could influence lipid metabolism and receptor dynamics in the brain [2].

A natural time-dependent progression in gene and protein expression patterns could be occurring independently of treatment, reflecting normal plasticity processes in the studied brain regions. 

To account for these effects, we have carefully considered statistical comparisons, ensuring that drug-induced effects are interpreted relative to their respective vehicle controls in each phase. These findings highlight the importance of understanding baseline shifts over time and suggest that vehicle-treated animals undergo their own adaptive responses to repeated procedures, which should be acknowledged when evaluating drug-specific effects.

1. Cloutier, S.; Newberry, R.C. Use of a conditioning technique to reduce stress associated with repeated intra-peritoneal injections in laboratory rats. Applied Animal Behaviour Science. 2008, 112, 158-173.

2. Szabo, Gy.; Dolganiuc, A.; Dai, Q.; Pruett, S.B. TLR4, Ethanol, and Lipid Rafts: A New Mechanism of Ethanol Action with Implications for other Receptor-Mediated Effects. The Journal of Immunology. 2007, 178, 1243-1249

Updated text in Experimental design:

“During the withdrawal period, animals were not handled or injected.” page 4, lines 143-144

Updated text in Discussion:

"Interestingly, significant alterations in gene and protein expression were also observed in the vehicle group during the withdrawal phase compared to the repeated injection phase. This suggests that repeated exposure to handling and injections may induce lasting neurobiological changes, even in the absence of pharmacological treatment. Additionally, cessation of vehicle administration, which contained ethanol and emulphor, might have contributed to observed differences, as vehicle components can influence lipid homeostasis and receptor regulation. These findings underscore the necessity of considering procedural effects when interpreting drug-induced changes." page 16, 559-567

Comment 2.6:

line 258: the section title is misleading because some markers went up in either the repeat administration or post-administration phases, in contrast to the title that suggests that both drugs "decrease markers of gliosis and inflammatory factors." This simplification also appears later (line 277-278) where it is stated definitively that there is an "immunosuppressive response" that isn't shown (were the immune systems of the rats actually challenged at any time?).

Response 2.6:

Thank you for the careful attention to the accuracy of our interpretation. To more accurately reflect our findings, we have revised the title.

Updated text:

“Repeated administration and withdrawal of WIN 55,212-2 and HU-210 modulate gliosis and inflammatory markers in the prefrontal cortex” on page 7, lines 267-268

Regarding the statement about an 'immunosuppressive response,' we agree that this wording was too conclusive. Instead, our findings suggest a complex regulatory effect on inflammatory and glial markers, with some markers decreasing while others increasing. Therefore, we have revised the statement (line 277-278) to more accurately reflect our results:

Updated text:

“Our findings indicate that repeated administration and withdrawal of WIN 55,212-2 modulate immune-related markers in the PFC, consistent with the role of endocannabinoids in regulating immune responses through cannabinoid receptors [8].” on page 8, lines 286-289.

These revisions aim to provide a more precise and balanced interpretation of our findings.

Comment 2.7:

3.5 (starting line 361): It is not clear that there is any hypothesis being tested in this section; this is dredging through data in search of finding correlations. This section, and the associated figure and SI tables, add nothing to the story and need to be removed.

Response 2.7:

We value the reviewer’s input regarding the interpretation of the correlation analyses. While we acknowledge that correlation alone does not imply causation, we believe these analyses provide valuable insight into the relationship between inflammatory markers and the endocannabinoid system during repeated administration and withdrawal of WIN 55,212-2 and HU-210. 

The rationale for including this section is based on previous evidence suggesting that endocannabinoid signaling plays a key role in modulating neuroinflammatory responses [1,2,3]. Given the observed changes in inflammatory and gliosis markers, we aimed to investigate whether these changes were associated with alterations in cannabinoid receptor and enzyme expression. These associations help frame the broader impact of synthetic cannabinoid exposure on neuroimmune interactions, which is a critical aspect of understanding their effects on the brain. 

To strengthen this section and better align it with the study’s overall hypotheses, we have now explicitly stated our a priori assumption that cannabinoid receptor and enzyme expression would be related to inflammatory responses, based on their known roles in immune regulation. Additionally, we have refined the discussion to emphasize that these findings are exploratory and should be interpreted within the context of existing literature. 

However, if the editorial team deems this section to be outside the scope of the main manuscript, we are open to moving it to the supplementary material.

1. Kasatkina, L.A.; Rittchen, S.; Sturm, E.M. Neuroprotective and Immunomodulatory Action of the Endocannabinoid System under Neuroinflammation. International Journal of Molecular Sciences, 2021, 22, 5431

2. Young, A.P.; Denovan-Wright, E.M. The Dynamic Role of Microglia and the Endocannabinoid System in Neuroinflammation. Frontiers in Pharmacology, 2022, 12, 806417

3. Camberos-Barraza, J.; Camacho-Zamora, A.; Bátiz-Beltrán, JC.; Osuna-Ramos, J.F.; Rábago-Monzón, A.R.; Valdez-Flores, M.A.; Angulo-Rojo, C.E.; Guadrón-Llanos, A.M.; Picos-Cárdenas, V.J.; Calderón-Zamora, L.; Norzagaray-Valenzuela, C.D.; Cárdenas-Torres, F.I.; De la Herrán-Arita, A.K. Sleep, Glial Function, and the Endocannabinoid System: Implications for Neuroinflammation and Sleep Disorders. International Journal of Molecular Sciences, 2024, 25, 3160

Updated text:

lines 368-400

“3.5. Relationship Between Gliosis, Inflammation Markers, and Endocannabinoid System During Repeated Administration and Withdrawal of WIN 55,212-2 and HU-210

Given the known interactions between the endocannabinoid system and neuroimmune signaling, we investigated whether gene expression changes in gliosis and inflammatory markers were associated with alterations in cannabinoid receptors and enzymes involved in endocannabinoid metabolism. We performed Spearman correlation analyses to explore these relationships.

During repeated administration in the PFC, Gfap, Ikbkb, and Nos2 expression levels showed a positive correlation with Cnr1 expression, despite no significant changes in expression being previously observed (Table S11) (Figure 6A). Additionally, Ikbkb, Nos2, Ptgs2, and Rela were positively associated with enzymes involved in 2-AG metabolism, including Dagl and Mgll (Table S11) (Figure 6A). These results suggest a potential link between inflammatory signaling and endocannabinoid metabolism during cannabinoid exposure.

Following withdrawal, inflammatory responses in the PFC appeared less pronounced, with positive correlations observed between Iba1 (which was reduced during WIN 55,212-2 and HU-210 withdrawal), Fcgr2b, and Cnr2 expression. Additionally, Ptgs2 and Rela were associated with Cnr1 expression (Table S12) (Figure 6B), indicating a persistent, yet altered, relationship between neuroinflammation and cannabinoid receptor signaling during withdrawal.

In the hippocampus, repeated administration of WIN 55,212-2 and HU-210 was associated with significant correlations between gliosis and inflammatory markers (Iba1, Gfap, Mrc1, Fcgr2b, Ikbkb, and Ptgs2) and components of the endocannabinoid system (Cnr1, Dagla, Daglb, Napepld, and Faah), suggesting that cannabinoid-induced neuroimmune modulation extends beyond the PFC (Table S13) (Figure 6C). However, during withdrawal, only a subset of these associations persisted, including correlations between Gfap, Il1b, and Ikbkb with Mgll, as well as Mrc1, Fcgr2b, and Il1b with Napepld and Faah (Table S14) (Figure 6D).

These findings support the hypothesis that synthetic cannabinoid exposure modulates neuroinflammatory responses in a phase-dependent manner, with withdrawal leading to partial resolution of these effects. While correlation analyses do not establish causation, they highlight key molecular relationships that warrant further mechanistic investigation into the immunomodulatory effects of cannabinoid receptor activation.”

Comment 2.8.1:

line 417 (and maybe elsewhere; lots of this in the paragraph starting line 469): it is definitively stated that there are "decreased endocannabinoid levels," which appears to be based on the observation that the expression levels of certain catabolic enzymes were altered; however, it does not appear that endocannabinoid levels were, indeed, determined. In the absence of this confirmation, one cannot make this definitive statement. At best, this language can be softened to say, "if endocannabinoid levels were altered..." and to include a clarifying statement that such levels were not, in fact, determined.

Comment 2.8.2:

Same comments in the discussion (469). Lots of commentary speculating what might happen if endocannabinoid levels were altered, but no confirmation that they were. In the absence of this confirmation, all of this is not very convincing of a mechanism.

Response 2.8.1&2:

We recognize the concern regarding our interpretation of endocannabinoid levels. In response, we have modified the text to clarify that our findings suggest potential alterations in endocannabinoid levels based on changes in the expression of their metabolic enzymes. We have also explicitly stated that direct quantification of endocannabinoid levels was not performed. The revised manuscript now includes more cautious phrasing, as well as a statement acknowledging the need for future studies to directly assess endocannabinoid concentrations in order to confirm our proposed mechanism.

Updated text:

“Interestingly, repeated administration of WIN 55,212-2 was associated with a decrease in protein levels of the glial marker IBA1 and inflammatory factors NF-κβ and IKKβ. Additionally, CB2, DAGLβ, and NAPE-PLD levels decreased, while FAAH increased. Both WIN 55,212-2 and HU-210 were also associated with decreased CB1 and MAGL levels and increased PPARα levels. page 14, lines 428-432

“If endocannabinoid levels were altered, this could be related to changes in cannabinoid-metabolizing enzymes, which are likely a result of increased exogenous cannabinoid concentrations reaching the PFC, and promoting an overactivation that leads to a compensatory decrease in cannabinoid receptor protein levels and desensitization of CB-mediated G protein activation [26-28]” page 14, lines 432-437

“Withdrawal of WIN 55,212-2 and HU-210 also led to a decrease in IBA1, GFAP, and the expression of inflammatory markers Fcgr2b and Rela, which could be indicative of a resolution phase.” page 15, lines 504-506

“Altogether, the observed decline in gliosis markers may reflect a resolution phase following the removal of synthetic cannabinoids. While changes in cannabinoid-metabolizing enzymes suggest a potential disruption or downregulation of endocannabinoid signaling, it is important to note that we did not directly measure endocannabinoid levels in this study. Rather, potential changes are inferred based on the observed expression patterns of their metabolic enzymes. Although our findings suggest a potential shift in endocannabinoid signaling, further studies are needed to directly quantify endocannabinoid levels and validate this hypothesis." page 15, lines 520-527

Comment 2.9:

lines 426-433: there is a lot of unfounded speculation into mechanisms in this section. For example, why would CB2 agonism be behind WIN's immunosuppressive effects when 1) that was not tested using a CB2 antagonist, 2) HU - which is also a CB1/CB2 agonist - did not cause the same effect, and 3) there are potentially many other mechanisms (line 449)? These speculative comments need to be reexamined in light of the fact that these studies did not attempt to connect CB1 or CB2 activity to the observed effects.

Response 2.9:

We consider the concern regarding our interpretation of CB2-mediated effects. We acknowledge that our study did not directly assess CB1 or CB2 receptor activity through pharmacological blockade, and therefore, our conclusions regarding CB2 involvement should be presented more cautiously and with limitation statements. In response, we have revised the text to reflect this uncertainty and highlight the possibility of alternative mechanisms.

Updated texts:

“The fact that only WIN 55,212-2, which reduced CB2 protein levels, was able to reduce IBA1, a marker of glial cells, and the inflammatory pathway of NF-κB and its activating molecule IKKβ, suggests that CB2-related mechanisms may play a role in the immunosuppressive effects of WIN 55,212-2. However, given that HU-210, another CB1/CB2 agonist, did not exert the same effect, and that CB2 involvement was not directly tested using an antagonist, alternative mechanisms cannot be ruled out. It is important to note that we did not perform pharmacological blockade of CB1 or CB2 receptors; thus, our findings do not confirm a direct causal relationship between CB2 activation and immunosuppression.” page 14, lines 440-448

“The differential effects of WIN 55,212-2 and HU-210 on immunosuppression in the PFC could be related to differences in their receptor binding profiles. HU-210 has a higher affinity and agonistic activity at CB1, which may influence glial activation differently. However, given the complexity of cannabinoid signaling, other receptor-mediated or downstream signaling mechanisms could also be involved.” page 14, lines 450-455

Comment 2.10:

SI: the two-way ANOVA interaction tables (that include interaction, addiction, drug) are confusing. It is not clear what we are looking at.

Response 2.9:

We have re-organized the statistical data of the supplementary tables in order to be clearer.
